# Post-Stroke Timing of ECM Hydrogel Implantation Affects Biodegradation and Tissue Restoration

**DOI:** 10.3390/ijms222111372

**Published:** 2021-10-21

**Authors:** Corina Damian, Harmanvir Ghuman, Carrinton Mauney, Reem Azar, Janina Reinartz, Stephen F. Badylak, Michel Modo

**Affiliations:** 1Department of Neuroscience, University of Pittsburgh, Pittsburgh, PA 15260, USA; COD41@pitt.edu (C.D.); CAM354@pitt.edu (C.M.); 2Department of Bioengineering, University of Pittsburgh, Pittsburgh, PA 15260, USA; Harmanvir.Ghuman@ucsf.edu (H.G.); REA54@pitt.edu (R.A.); badysx@UPMC.EDU (S.F.B.); 3McGowan Institute for Regenerative Medicine, University of Pittsburgh, Pittsburgh, PA 15260, USA; 4Department of Radiology, University of Pittsburgh, Pittsburgh, PA 15260, USA; jreinartz97@web.de; 5Department of Surgery, University of Pittsburgh, Pittsburgh, PA 15213, USA

**Keywords:** extracellular matrix, hydrogel, therapeutic window, stroke, regeneration, tissue repair, biodegradation, biomaterial, scaffold, cell invasion

## Abstract

Extracellular matrix (ECM) hydrogel promotes tissue regeneration in lesion cavities after stroke. However, a bioscaffold’s regenerative potential needs to be considered in the context of the evolving pathological environment caused by a stroke. To evaluate this key issue in rats, ECM hydrogel was delivered to the lesion core/cavity at 7-, 14-, 28-, and 90-days post-stroke. Due to a lack of tissue cavitation 7-days post-stroke, implantation of ECM hydrogel did not achieve a sufficient volume and distribution to warrant comparison with the other time points. Biodegradation of ECM hydrogel implanted 14- and 28-days post-stroke were efficiently (80%) degraded by 14-days post-bioscaffold implantation, whereas implantation 90-days post-stroke revealed only a 60% decrease. Macrophage invasion was robust at 14- and 28-days post-stroke but reduced in the 90-days post-stroke condition. The pro-inflammation (M1) and pro-repair (M2) phenotype ratios were equivalent at all time points, suggesting that the pathological environment determines macrophage invasion, whereas ECM hydrogel defines their polarization. Neural cells (neural progenitors, neurons, astrocytes, oligodendrocytes) were found at all time points, but a 90-days post-stroke implantation resulted in reduced densities of mature phenotypes. Brain tissue restoration is therefore dependent on an efficient delivery of a bioscaffold to a tissue cavity, with 28-days post-stroke producing the most efficient biodegradation and tissue regeneration, whereas by 90-days post-stroke, these effects are significantly reduced. Improving our understanding of how the pathological environment influences biodegradation and the tissue restoration process is hence essential to devise engineering strategies that could extend the therapeutic window for bioscaffolds to repair the damaged brain.

## 1. Introduction

The intrinsic ability of tissues and organs to repair and regenerate varies tremendously [1]. The brain, especially, is considered to have a very limited capacity for regeneration of lost tissue [2]. This severely restricts the potential for recovery after acute tissue injuries, such as stroke [3]. Although transplantation of neural stem cells can replace some lost neurons in peri-infarct tissues following stroke, it does not replace lost tissues [4]. MRI scanning in stroke patients shows that 94% had tissue loss (i.e., cavitation), which was incomplete at 30 days following infarction but almost always present at 90 days [5]. In preclinical studies, a tissue cavity can typically be observed 10 days post-stroke, but extracellular matrix (ECM) is still present within the lesion core up to 7–10 days post-stroke [6]. In rats, this tissue cavity can increase up to 4 weeks after a stroke [7].

Tissue cavitation is part of the pro-inflammatory phase after a stroke and is characterized by the infiltration of immune cells, such as macrophages. Microglia/macrophages phagocytose cellular debris and secrete pro-inflammatory cytokines that affect surrounding tissue [8]. Pro-inflammatory M1-like macrophages are found within the lesion core, degrading the remaining tissue ECM by secretion of proteases, such as matrix metalloproteinases (MMPs), prior to phagocytosis [9]. This secondary damage progresses over time to define a tissue cavity [10]. Once cavitation is complete, the level of MMPs decreases, and the environment stabilizes with a glial scar sealing off viable tissue [11] from cytotoxic extracellular fluid (ECF) filling the cavity [5]. The glial scar is a consequence of the peri-infarct inflammatory response that produces morphological and functional changes in reactive astrocytes [12]. Compacting astrocytes at the edge of the lesion core produces a tightly packed hyper-filamentous astrocyte border [13] that intensifies as the cavity forms. Deposition of glycosaminoglycans (GAGs), such as chondroitin sulfate, and hyaluronan (HA), solidify hypertrophic astrocytes into a scar that inhibits axonal regeneration and remyelination [14]. The formation of the glial scar signals the end of the central nervous system injury response, which can take several weeks to fully complete [14].

The glial scar also releases factors that mediate the tissue inflammatory response and subsequent remodeling of the lesion [15,16]. This remodeling process is characterized by a shift from a pro-inflammatory to a pro-repair phase, which persists for several months following a stroke in the peri-infarct tissue [17,18]. This shift is very apparent in macrophages, with a high proportion (>35%) exhibiting pro-inflammatory (M1) characteristics (i.e., the release of MMPs, phagocytosis) immediately following a stroke, but by 14-days mostly (>20%) pro-repair (M2) macrophages contribute to processes, such as angiogenesis and synaptic pruning [19]. The host response to implanted ECM hydrogel bioscaffolds mimics this inflammatory cascade and is hence characterized by the infiltration of an initial pro-inflammatory cell population followed by a macrophage polarization toward an M2-like phenotype [19,20,21,22,23]. As biodegradation of the ECM hydrogel is essential to effect tissue regeneration, the presence of M1-like macrophages is essential to secrete proteases that break down the cross-linked collagen within these hydrogel bioscaffolds [24,25,26]. This process slowly removes structural elements of the scaffold that guide the invasion of host cells [20]. A variety of other cell types, including endothelial cells, neural progenitor cells, and astrocytic cells, eventually produce their own ECM leading to the formation of new brain tissue inside the lesion cavity [22].

The therapeutic potential of ECM hydrogel needs to be considered in the context of this evolving post-injury environment. It currently remains unclear if tissue cavitation is required to afford the implantation of a volumetric bioscaffold. It is also important to establish if a pro-inflammatory environment, characterized by the presence of macrophages and MMPs, is advantageous to initiate the biodegradation of an ECM hydrogel. We, therefore, hypothesize that biodegradation and tissue regeneration will be decreased in the absence of ongoing inflammatory activity in the peri-infarct area. Consequently, post-stroke timing of implantation of an ECM hydrogel will impact its potential to induce tissue regeneration. The objective of the present study was hence to determine how post-stroke implantation timing (7-, 14-, 28-, and 90-days) of an ECM hydrogel into a stroke cavity affects its delivery, biodegradation, host response, and tissue regeneration.

## 2. Results

### 2.1. Biodegradation Is Influenced by Post-Stroke Implantation Timing

The host tissue’s response to acute damage changes over time, thereby affecting its ability to degrade ECM bioscaffolds implanted into a tissue cavity. Implantation of escalating volumes (50, 100 150 μL) of a 4 mg/mL ECM hydrogel at 7-days post-stroke did not result in an in situ bioscaffold volume greater than 22 µL due to a lack of cavitation (Figure 1). The presence of tissue remnants, especially non-degraded ECM, prevented the delivery of bioscaffold volumes required to cover the lesion area. The formation of a tissue cavity was hence essential to deliver the ECM hydrogel for tissue restoration. By 14-days post-stroke, tissue cavitation was extensive and afforded the implantation of ECM hydrogel, providing the earliest therapeutic time point. A robust bioscaffold implantation occurred at 14-, 28- and 90-days post-stroke (Figure 2A). Glial scarring delineating the tissue cavity and degrading bioscaffolds was evident 14-days post-implantation, with the formation of new tissue evident in the 14- and 28-days post-stroke conditions (Figure 2B). 

Lesion volumes between animals that received ECM hydrogel and MCAo only rats were equivalent for all post-stroke implantation time points (Figure 2C). The amount of ECM hydrogel remaining was equivalent in the 14- and 28-days post-stroke implantation groups (n.s.) but significantly lower than in the 90-days implantation group (F = 17.78, *p* < 0.001) (Figure 2D). Up to 93%, with an average of 86%, of ECM hydrogel was degraded when implanted 28-days post-stroke. Although 14-days post-stroke implantation effected an average of 70% degradation, this was not significantly different from the 28-days post-stroke implantation time point. In contrast, the lower (64%) degradation efficiency observed after 90-days post-stroke implantation was significantly reduced (F = 26.76, *p* < 0.0001, Figure 2E) compared to both 14- (*p* < 0.001) and 28-days post-stroke implantation (*p* < 0.001). However, the daily rate of biodegradation (μL/day) was equivalent between all implantation time points (F = 0.9123, n.s.). These results indicate that a tissue cavity is a *conditio sine qua non* to deliver ECM bioscaffolds for the purpose of tissue regeneration and that earlier time points effect greater biodegradation. 

### 2.2. Post-Stroke Maturation of Glial Scar Affects Biodegradation 

A mature glial scar at 90-days post-stroke provides a physical boundary between the host tissue and the ECM bioscaffold (Figure 3A), whereas at earlier time points astrocyte density of this tissue boundary is less compact (Figure 3B). The presence of a mature glial scar creates a distinct division between the host tissue and the ECM bioscaffold. Quantification of the glial scarring indicates that GFAP intensity was significantly higher at 90-days post-stroke than at 14- (*p* < 0.01) or 28-days post-stroke (*p* < 0.05) (Figure 3C). GFAP intensity is maintained at a moderate level throughout the peri-infarct tissue 14-days post-stroke, indicating an ongoing process of astrocytosis. In contrast, by 28-days post-stroke, GFAP intensity increases at the tissue boundary, with astrocytosis in the peri-infarct tissue decreasing. The change in GFAP reactivity is further amplified after 90-days with a mature and intense glial scar at the tissue boundary and low GFAP reactivity in the surrounding tissue. The maturation of the glial scar at the tissue boundary is hence a potentially important co-variate affecting the interaction of the bioscaffold with veterate host tissue (i.e., tissue present before ECM implantation).

### 2.3. Post-Stroke Implantation Timing Affects Host Cell Infiltration

Cell infiltration into the bioscaffold is critically important for biodegradation and tissue regeneration. Host cells need to infiltrate the bioscaffold by crossing through the glial scar from the peri-infarct tissue (Figure 4A). There is a stark contrast in cells present within the bioscaffold depending on its post-stroke implantation time. In the 14- and 28-days post-stroke implantation conditions, a high density of cells are distributed throughout the bioscaffold, and a marked degradation of the ECM is evident in these regions (Figure 4B). In comparison, bioscaffolds implanted 90-days post-stroke are poorly degraded, and cells are more sparsely distributed. 

Quantification of DAPI+ cells inside the bioscaffold (Figure 4C) revealed that the total amount of cells invading the bioscaffold was the lowest in the 14-days post-stroke implantation condition (F = 1.952, n.s.), whereas the highest level (average of 117,056 cells) was observed in the 28-days post-stroke implantation condition (Figure 4D). Although the total number of cells in the bioscaffold in the 90-days post-stroke implantation condition was almost twice that of the 14-days condition, the larger ECM volume resulted in a significantly (F = 24.02, *p* < 0.001, post-hoc test *p* < 0.01) lower cell density in the 90-days post-stroke condition. 

The tissue/bioscaffold interface is a key factor in facilitating host cells to infiltrate the bioscaffold and effect biodegradation. In the case of diffuse scarring, as observed in the 14-days post-stroke implantation time point, individual host cells can infiltrate the bioscaffold without the hindrance of a dense scar formation, and no sharp boundary at the tissue/bioscaffold interface is observed (Figure 4E). Alongside this individual cell infiltration, collective cell migration, akin to that observed in cancer, can be observed to foray into the hydrogel (Figure 4D). These multicellular spears push into the bioscaffold while forming a macro track that remodels the surrounding ECM. Changes at the tissue/bioscaffold boundary, associated with glial scar maturation at different time points, are hence a major determinant of cell infiltration and density within the ECM hydrogel. 

### 2.4. Macrophage Invasion and Pro-Repair Polarization Is Reduced with a Delay in Post-Stroke Implantation

Macrophages are largely responsible for the biodegradation of ECM hydrogels and constitute a major fraction of infiltrating cells during the initial post-implantation phase. Macrophages accumulate extensively in the immediate peri-infarct tissue before invading and distributing throughout the bioscaffold (Figure 5A). Quantification of Iba1+ cells reveals that there is a low density of macrophages in the contralateral hemisphere across all implantation time points (F = 1.879, n.s., Figure 5B), which aligns with what would be expected in normal, uninjured tissue. In the peri-infarct region, macrophage density for the 28-days implantation time point is significantly (*p* < 0.05) greater than for 90-days post-stroke implantation. The peri-infarct density of macrophages was higher (*p* < 0.05) than for the contralateral hemisphere for the 28-days post-stroke implantation time point (F = 4.806, *p* < 0.05), but only marginally increased for the 14- (72%) and 90-days’ time point (49%). Within the ECM bioscaffold, almost double the macrophage density of the peri-infarct region was observed. Macrophage density in the 28-days post-stroke implantation group was significantly higher than for the 90-days post-stroke condition, although it was non-significantly different from the 14-days post-implantation condition. A lower macrophage density within bioscaffolds in the 90-days post-stroke implantation group is consistent with a lower cell infiltration and biodegradation. The lower cell density in the peri-infarct area, in contrast, indicates that a lower recruitment of macrophages occurred at this time point. 

Distinguishing pro-inflammatory M1-like (CD86+) and pro-repair M2-like (CD206+) macrophages revealed that there is a major shift between macrophages infiltrating the peri-infarct area, which are predominantly of an M1-like phenotype, versus those within the bioscaffold also exhibiting M2-like characteristics (Figure 6A). M1- and M2-like, as well as M1+M2 co-expressing, macrophages were quantified to determine the effect of post-stroke implantation timing upon macrophage phenotype and function (Figure 6B). In the contralateral hemisphere very low levels were observed for all macrophage phenotypes (M1: F = 0.7663, n.s.; M2: F = 0.01356, n.s.; M1 + M2: F = 0.3296, n.s., Figure 6C). In contrast, within the peri-infarct region M1-like macrophages were more than twice as common than M2 or M1+M2 co-expressing cells (M1: F = 7.604, *p* < 0.05; M2: F = 0.2687, n.s.; M1 + M2: F = 0.05726, n.s.). The cell density of M1-like macrophages was significantly (*p* < 0.05) higher in the 28-days post-stroke implantation condition, indicating a strong pro-inflammatory response. Within the bioscaffold, the density of M1- and M2-like cells was almost equivalent, with both phenotypes exhibiting the highest level in the 14-days post-stroke implantation time point. The 14-days post-stroke implantation time point also had the highest density of M1 + M2 co-expressing cells (M1: F = 5.429, *p* < 0.05, M2: F = 9.347, *p* < 0.01, M1 + M2: F = 3.630, n.s.). 

### 2.5. Bioscaffold Vascularization Is Reduced with a 90-Day Post-Stroke Implantation

Neovascularization is important for cell infiltration and for the long-term survival of these invading cells. At the 14- and 28-days time points, there was significant vascularization, with branching of different vessels inside the ECM hydrogel (Figure 7A). At the 90-days post-stroke time point, however, there was little vascularization observed. Neovascularization of the bioscaffold supports cell infiltration into the ECM hydrogel (Figure 7B), providing an alternative route to cell infiltration than through peri-infarct tissues. Quantification of ECs (RECA-1+) revealed that the contralateral (F = 1.049, n.s.) and peri-infarct (F = 0.9014, n.s.) regions have relatively consistent density of ECs across all time points (Figure 7C). Within the biomaterial, the 14- and 28-days conditions have a significantly (F = 10.94, *p* < 0.01) increased density of ECs compared to the 90-days post-stroke implantation (*p* < 0.05). The density of ECs in the bioscaffold implanted at 14- or 28-days post-stroke was more than twice that of the contralateral and peri-infarct regions, whereas in the 90-days post-stroke implantation condition, EC density was less than half the cell density of other regions. These findings indicate that the 90-days post-stroke implantations present an unfavorable time window for bioscaffold-induced neovascularization of the tissue cavity caused by a stroke.

### 2.6. Delayed Implantation of ECM Hydrogel Reduces Oligodendrocytes in the Bioscaffold

To restore brain tissue inside the lesion cavity, the infiltration of neural cells is essential. Re-colonization with neurons is dependent on the migration of neural progenitors. Neural progenitors (DCX+) are observed migrating into the ECM hydrogel through peri-infarct tissues directly into the bioscaffold. In the presence of blood vessels, such as the remains of the MCA, extensive migration along these blood vessels occurs (Figure 8A). Neural progenitor cells mature into neurons (NeuN+, Figure 8B), astrocytes (GFAP+, Figure 8C), and oligodendrocytes (CNPase+, Figure 8D) within the bioscaffold. 

A quantification of these phenotypes revealed that there was no significant difference in DCX+ neural progenitor density in the contralateral hemisphere (F = 0.1352, n.s.), the peri-infarct region (F = 2.807, n.s.) and within the ECM hydrogel (F = 4.742, *p* < 0.05) at any time point post-stroke (Figure 9). However, the DCX+ cell density in the ECM was significantly less for 14- (4× less, *p* < 0.05), but not 28- or 90-days post-stroke conditions. No statistical difference was evident for mature neurons (NeuN+) in the in the contralateral hemisphere (F = 0.5185, n.s.), peri-infarct region (F = 1.378, n.s.) or within the ECM bioscaffold (F = 3.735, n.s.) across any implantation time point. The contralateral measurements provide a reference of densities in undamaged tissue. It is evident here that at 14-days post-implantation, neuron densities within the ECM are significantly reduced (*p* < 0.05), constituting only 25% of neuron densities in the contralateral hemisphere across all time points. 

Astrocyte density in the contralateral hemisphere was not significantly different (F = 0.7385, n.s.) at any time point (Figure 8), but there was a significant decrease in the density of GFAP+ cells in the peri-infarct region at 90-days post-stroke (*p* < 0.05). Peri-infarct astrocyte density was significantly increased (F = 6.023, *p* > 0.05) at 14- (*p* < 0.05) and 28-days post-stroke (*p* < 0.01), but not 90-days post-stroke. Within the ECM, astrocyte density was significantly increased upon implantation 28-days post-stroke (F = 5.847, *p* < 0.05), compared to 14- (*p* < 0.05) and 90-days post-stroke (*p* < 0.05). Astrocyte density within the ECM was not significantly different from the contralateral hemisphere, but was significantly reduced compared to the 14-days peri-infarct region (*p* < 0.05).

The density of oligodendrocytes in the contralateral hemisphere was equivalent at all time points (F = 0.1963, n.s.), but in the peri-infarct region (F = 3.711, n.s.), the average density was twice as high at the 28-days post-stroke time point. This level of oligodendrocytes was also observed within the ECM for the 14- and 28-days post-stroke implantation time points. Oligodendrocyte density in these conditions was twice the level of the contralateral hemisphere. Within the ECM bioscaffold, the 90-days post-stroke implantation condition revealed a significantly (F = 6.213, *p* < 0.05) reduced oligodendrocyte density compared to both the 14- (*p* < 0.05) and 28-days implantation time points (*p* < 0.05). These results indicate that phenotypes reconstituting brain tissue are reduced in the bioscaffold if this is implanted 90-days post-stroke, whereas 28-days post-stroke is the most favorable time point for ECM hydrogel implantations. 

## 3. Discussion

ECM hydrogel implantation offers a new therapeutic paradigm for the treatment of stroke by inducing brain tissue regeneration within the tissue cavity [20]. We here demonstrated that the therapeutic window for this novel approach is dependent on a tissue cavity having formed to allow the delivery of the bioscaffold. An optimal therapeutic time window for bioscaffold-induced tissue regeneration occurs between 14-days post-stroke and extending beyond 28-days but is significantly reduced by 90-days post-stroke. These results highlight the importance of considering the post-stroke tissue environment and its interaction with a bioscaffold to ensure an effective tissue regeneration occurs. 

### 3.1. Impact of Post-Stroke Implantation Timing on Biodegradation

Small quantities of ECM hydrogel can be injected into damaged brain tissue to enhance neurological recovery [27,28,29]. However, large volumes of ECM hydrogel aimed at regenerating lost tissue require the formation of a tissue cavity that affords the injection of ECM pre-gel and drainage of superfluous extracellular fluid to vacate the space for bioscaffolding [30]. We here demonstrated that within 7 days after a stroke, cavitation is incomplete, and only a small volume of ECM hydrogel could be delivered to the lesion core. In contrast, 14 days post-stroke, the core of the lesion was cleared of both cells and tissue ECM, creating a tissue cavity that could accommodate the bioscaffold. Therefore, 14-days post-stroke was the earliest time point to efficiently and consistently implant an ECM hydrogel, but earlier interventions could be considered if appropriate image guidance is available to indicate that sufficient cavitation occurred [31]. 

It is important to note that a 2–3 weeks post-stroke time frame is also commonly considered a post-stroke period during which the immune response shifts from a pro-inflammatory to a pro-repair environment [32]. Microglia and macrophages are the main cell types implicated in the inflammatory response to stroke. Phagocytosis of cellular debris is affected by microglia/macrophages, as well as the production of matrix metalloproteinase (MMPs) [9] that degrade the remnant tissue ECM and cause a cavity to form [8,10]. After cavitation is complete, MMPs return to baseline levels, whilst microglia/macrophages shift to a pro-repair response [33,34]. In addition to the space required for the delivery of a bioscaffold, the 14-days post-stroke time point also provides potentially a favorable inflammatory environment to initiate the biodegradation of an ECM hydrogel. We hypothesize that the presence of MMPs in the tissue cavity, released by peri-infarct cells, can initiate biodegradation of the ECM hydrogel. The degradation of the bioscaffold releases chemoattractant factors that promote macrophage infiltration into the brain and the bioscaffold. The degradation products of ECM have repeatedly been shown to mitigate the M1-like macrophage phenotype and promote an increased M2/M1 macrophage ratio [21,35,36,37]. ECM hydrogel implanted at 14- and 28-days post-stroke revealed efficient biodegradation, whereas biodegradation of bioscaffolds implanted 90-days post-stroke was significantly reduced. A key question that emerges is, therefore, how levels of MMPs in the tissue cavity affect the biodegradation cascade and whether macrophages can infiltrate an ECM hydrogel in the absence of a host MMP-induced biodegradation.

Implantation of the ECM bioscaffold replicates the pro-inflammatory to pro-repair response that is characteristic of post-stroke host tissue [19,20,21,22,23]. In host tissue, this shift is apparent in the polarization of macrophages, with a high proportion of M1-like macrophages immediately after stroke followed by an increase in M2-like macrophages by 14-days after a stroke [19]. After 14-days post-implantation, all the injection time points revealed an equal ratio of M1- to M2-like macrophages, although the tissue density of macrophages gradually decreased post-stroke. This indicates that the magnitude of the inflammatory response is regulated by the post-stroke timing, whereas the polarization of the response is determined by the ECM hydrogel. It can therefore be concluded that the efficiency of the biodegradation is governed by the microenvironmental conditions at the implantation site, whereas the type of the inflammatory response is regulated by the biochemical and biophysical properties of the bioscaffold. 

The infiltration of cells from the host tissue into the ECM hydrogel is potentially limited by the formation of a glial scar that gradually forms after a stroke to seal off the tissue cavity from the host tissue. During the process of tissue cavitation, reactive astrocytes proliferate in viable tissue, gradually defining the cavity [11]. By 14-days post-stroke, this tissue edge, characterized by proliferating and reactive astrocytes, remains rather mesh-like, with macrophages migrating through this microenvironment to remove ECM remnants [38]. Infiltration of macrophages into the ECM hydrogel at this time point, therefore, is not much impacted by the microenvironment at the edge of viable brain tissue. However, as shown here, this microenvironment compacts and matures into a glial scar by 90-days post-stroke [14]. Glial scarring in a chronic stroke environment could hence significantly impede the migration of macrophages (and other cells) into the bioscaffold. Still, it remains unclear at what time point between 28- and 90-days post-stroke maturational changes occur in the glial scar that would shift the endogenous tissue response from a permissible to a prohibitive milieu.

### 3.2. Brain Cell Invasion Is Reduced with Prolonged Post-Stroke Implantation

It is possible that immune cell invasion is stimulated by ECM pre-gel or MMP-degraded hydrogel permeating into the peri-infarct area. A mature glial scar could reduce or even eliminate these molecules to enter the veterate brain and induce a host response. It currently remains unclear if a host response to the bioscaffold is dependent on ECM molecules from the bioscaffold permeating into the veterate brain or if, for instance, the pressure of the implantation onto the veterate brain could invoke an initial immune response that then initiates the biodegradation process. Glial scars have been found to be permeable to some small molecules while still acting as a robust physical barrier to cell migration [9]. More detailed mechanistic studies are hence required to improve our understanding of how ECM hydrogels interact with the different veterate microenvironments encountered post-stroke. Nevertheless, the maturation of the glial scar here coincides with reduced biodegradation and cellular infiltration, indicating that the tissue-bioscaffold interface plays a major role in defining the therapeutic window of bioscaffold-induced tissue regeneration. 

Not only is the infiltration of macrophages pivotal for biodegradation, but neovascularization of the bioscaffold is another critical step in de novo tissue formation [19,39]. Neovascularization occurred through the infiltration of endothelial cells branching off from the existing vasculature in peri-infarct tissues, as well as potential remnants of the MCA. As in previous studies [19], ECM hydrogel induced an efficient and extensive vascularization of the lesion cavity if implantation occurred 14- or 28-days post-stroke, but poor neovascularization was observed with a 90-days post-stroke implantation paradigm. Glial scarring and inefficient macrophage infiltration at 90-days post-stroke potentially impede the neovascularization process. In contrast, the efficient neovascularization at earlier time points provides a conduit for neural cells to infiltrate the bioscaffold, as well as for macrophages to directly infiltrate into the bioscaffold without having to migrate through peri-infarct tissues. Neovascularization of the ECM hydrogel hence could accelerate biodegradation and support the constructive remodeling of the bioscaffold. 

The coupling of angiogenesis and neurogenesis has been extensively demonstrated during brain development, cancer, as well as in transplantation paradigms [40]. To form brain tissue, the infiltration of neural cells is required [39]. Neural progenitor cells invade the bioscaffold through chain cell migration, with “leader cells” defining a migration path [41]. However, we also here observed migration of neural progenitors along blood vessels before diverting into the bioscaffold for terminal differentiation. Neural progenitor cells, as well as neurons, astrocytes, and oligodendrocytes were present within the degrading bioscaffold, regardless of post-stroke implantation timing. However, cell densities at the 90-days post-stroke implantation time point were significantly reduced for mature phenotypes, suggesting an impaired tissue restoration process. Even in the 14- and 28-days post-implantation time point conditions, neurons were approximately 8× fewer than in the intact contralateral hemisphere. Although a continued infiltration of neural progenitors and longer maturation time can further improve neuron density in the de novo tissue, it is likely that an enhancement of endogenous neurogenesis is required to promote a more robust de novo brain tissue formation. 

During stroke recovery, there is an increase in oligodendrocyte progenitor cells (OPCs) in the peri-infarct area, some of which mature into myelinating oligodendrocytes [42,43,44]. High levels of CNPase+ differentiating OPCs were observed in both the peri-infarct region and within the hydrogel in the 14- and 28-days post-stroke implantation condition, indicating a robust peri-infarct OPC proliferation and hydrogel invasion. In contrast, implantation at 90-days post-stroke revealed a much reduced OPC density in the peri-infarct area and almost no OPCs in the bioscaffold. It remains unclear what factors regulate oligodendrogenesis post-stroke [45], as well as within the hydrogel, but OPCs are essential to myelinate sprouting axons [43,46], which will be required to form a de novo tissue that is interconnected with the veterate brain.

### 3.3. Translational Considerations for a Therapeutic Window of Bioscaffold Implantation

ECM hydrogel concentration affects its biodegradation and determines its potential to induce brain tissue regeneration [30]. Most of the ECM is degraded by 14-days post-implantation and is indicative of long-term outcome, but even at 90-days post-implantation, the process of tissue restoration is not complete [19]. We here demonstrated further that the timing of implantation post-stroke affects this process. A therapeutic time window starting at 14-days post-stroke, peaking around 28-days post-stroke, and being significantly decreased by 90-days suggests that implantation of ECM hydrogel is most efficacious for brain tissue restoration if implanted 1-month post-stroke. Longer-term studies will still need to establish the time course of a complete tissue restoration and if this will support functional and behavioral recovery. 

It is, however, conceivable that the therapeutic window of ECM hydrogel can be further extended by introducing additional bioengineering strategies that would, for instance, reduce glial scarring [47,48] and/or control the levels of MMPs in the tissue cavity. Establishing the specific levels of tissue cavitation and glial scaring through MRI, as well as MMP levels through biopsy, could hence provide greater control over bioscaffold-induced tissue regeneration. In a translational setting, gaining control over these factors would be important to ensure optimal therapeutic efficacy for each individual patient. Advances are being made in developing, for instance, MRI-based measurements of bioscaffold distribution and degradation [49,50,51], as well as monitoring the infiltration of macrophages [52]. Diffusion-based MRI might further report on the tissue restoration process [53] and provide a means to evaluate mesoscale connectivity between different brain regions [54]. It is important to elucidate these 1st and 2nd order effects of bioscaffolds’ interaction with host tissue to ensure the therapeutic application of these for the treatment of stroke.

It currently remains unclear, for instance, what level of tissue restoration would be required to impact behavioral impairments after a stroke. We previously demonstrated that poor biodegradation of ECM hydrogel in the stroke cavity fails to restore a de novo tissue and hence did not impact behavioral impairments [39], but the long-term development of de novo tissue under optimal therapeutic conditions remains unknown. Urinary bladder matrix (UBM) ECM has been extensively characterized, and some of its molecular signaling involved in tissue restoration is being dissected, but the specific desirable molecular composition of inductive bioscaffolds remains unknown [55]. Ideally, the bottom-up design of hydrogels with different chemical and physical characteristics can exploit and improve bioscaffold-induced brain tissue regeneration for the treatment of stroke [56].

## 4. Materials and Methods

### 4.1. Extracellular Matrix (ECM) Hydrogel 

ECM hydrogel was produced via mechanical isolation of the basement membrane and tunica propria of porcine urinary bladder (Tissue Source Inc., Lafayette, IN). The bladder was subjected to a decellularization process in which the isolated layers were immersed in 0.1% peracetic acid in ethanol with agitation (4% *v*/*v*; 120 min; 300 rpm). To remove any remaining cellular debris, a series of rinses with PBS and deionized water was performed [57]. Decellularization was confirmed via Hematoxylin & Eosin, 4′,6-diamidino-2-phenylindole (DAPI) staining, agarose gel electrophoresis, and quantification of remnant DNA [58]. ECM was lyophilized, comminuted, and solubilized with pepsin (1 mg/mL) in 0.01 N HCl. ECM powder was stored at 4 °C. Formulation of the ECM powder as a hydrogel was achieved by dilution to the desired concentration of 4 mg/mL with PBS [57], while pH neutralization was achieved by adding 0.1 N NaOH. This preparation had a viscosity of 0.084 Pa s, with a storage modulus (G’) of 76.6 Pa and a loss modulus (G”) of 11.0 Pa [30].

The ECM material is predominantly (~70%) composed of collagen [59] but also contains other prominent ECM proteins, including decorin, fibronectin, and laminin subunit γ1 [60]. A variety of different growth factors known to influence neuronal and endothelial cells, such as vascular endothelial growth factor-A, transforming growth factor -β, nerve growth factor, and basic fibroblast growth factor, are retained during ECM preparation. Matrix bound nanovesicles (MBV) with abundant miRNA and other signaling molecules, including immunomodulatory signaling molecules, are also present within the ECM preparation [61]. 

### 4.2. Middle Cerebral Artery Occlusion (MCAo)

All animal procedures complied with the US Animals Welfare Act (2010) and were approved by the University of Pittsburgh Institutional Animal Care and Use Committee (IACUC, Protocol 14051233, approval date 1 May 2014). Sprague-Dawley rats (12 weeks old males, 260 ± 15 g, Taconic Labs, Rensselaer, NY) were kept on a 12-h light/dark schedule, with food and water available ad libitum. To induce an ischemic stroke in rats, a transient intraluminal right middle cerebral artery (MCA) occlusion was applied using a 5-0 silicone rubber-coated monofilament (diameter 0.12 mm, length 30 mm, tip coating at 0.35 mm for 5–6 mm, Doccol, Sharon, MA). While the animal was under isoflurane anesthesia (4% induction, 1% maintenance in 30% O_2_), the monofilament was inserted into the opening of the MCA in the circle of Willis [62]. The MCA was occluded for 70 min prior to reperfusion by retracting the filament to the common carotid bifurcation. This produced an occlusion in rats that is most similar to 2/3 of all cases of human ischemic stroke [63], which involve the MCA [64]. Rats were assessed for forelimb flexion and contralateral circling after recovering from the anesthesia. Post-surgical pain was alleviated with the analgesic buprenorphine (0.05 mg/kg, i.p.) twice daily for 3 days. Daily post-operative care and neurological assessments were performed until rats recovered pre-operative weight [62,65]. Animals not exhibiting signs of MCA damage or who failed to recover weight were excluded from the study [39].

### 4.3. Magnetic Resonance Imaging (MRI) 

#### 4.3.1. Acquisition

Ten days after the MCAo, rats were anesthetized with isoflurane (4% induction, 1% maintenance in 30% O_2_) to assess the presence, location, and volume of tissue loss using a T_2_-weighted spin-echo MRI sequence (TR = 6000 ms, TE = 8 ms, 8 Averages, FOV 30 × 30 mm, 128 × 128 matrix, 42 slices at 0.5 mm thickness) on a horizontal bore 9.4 T Varian scanner [30,31]. 

#### 4.3.2. Lesion Volume and Intensity Measurements

Stroke damage was determined by a hyperintense signal on T_2_-weighted images that was thresholded at 1 standard deviation above the mean of a rectangular region of interest (ROI) in the contralateral hemisphere, encompassing striatum, corpus callosum, and neocortex [6,66]. Two animals were excluded due to insufficient damage. Rats (n = 28) with a lesion volume >40 mm^3^ (i.e., 40 μL) were randomly assigned into either untreated or ECM treated groups, resulting in an equivalent distribution of lesion volumes (range: 40–250 mm^3^) across all groups. 

### 4.4. MRI-Guided Implantation of ECM Hydrogel

Rats underwent either ECM hydrogel implantation or PBS injection at 7-, 14-, 28-, and 90-days post-stroke (Table 1). The rationale for choosing these post-stroke time points was based on the evolving lesion environment: 7 days (sub-acute stroke)—ongoing cavitation with tissue ECM mostly remaining intact, ongoing pro-inflammatory environment, no glial scarring; 14 days (sub-acute stroke)—cavitation complete, inflammatory response switching to pro-repair, signs of early glial scarring; 28 days (sub-chronic stroke)—maturing glial scar, maximal pro-repair inflammatory response; 90 days (chronic stroke)—chronic lesion environment with mature glial scarring surrounding cavitated tissue, negligible inflammatory response. Implantation was achieved by placement into a stereotactic frame (Kopf, Tujunga, CA) under isoflurane anesthesia (4% induction, 1% maintenance in 30% O_2_). A vertical skin incision exposed Bregma on the skull, which acted as a marker for the location of two burr holes in relation to the T_2_-weighted MR image [30,31]. The first burr hole afforded the administration of the 4 mg/mL ECM hydrogel through a 250 μL Hamilton syringe with a 24 G beveled metal tip needle (Hamilton, Reno, NV) into the lower quarter of the lesion core. A drainage cannula (24 G) was placed at the topmost position of the tissue cavity within the second burr hole to evacuate extracellular fluid from the lesion cyst [30]. ECM hydrogel was injected (10 μL/min) using a frame-mounted injection pump (World Precision Instruments, Sarasota, FL) to deliver a volume equivalent to that of the lesion core (41–250 μL). The injected ECM displaced and drained the less dense necrotic debris from the stroke cavity. After injection, the needle/cannula was left in place for 5 min to give time for the material to dissipate and gel in situ at 37 °C body temperature. For a 4 mg/mL ECM concentration, time to 50% gelation is 3.2 min [30]. Finally, burr holes were filled with bone wax (Fisher), and the incision was sutured. LMX4 (Ferndale Laboratories, Ferndale, MI, containing 4% Lidocaine) was topically applied as an analgesic, and buprenorphine (0.05 mg/kg i.p.) was administered daily for 3 days to provide sustained pain relief.

### 4.5. Histologic Analyses

#### 4.5.1. Perfusion-Fixation of Tissue

For analysis of the in situ distribution of ECM hydrogel and cell infiltration, rats were perfused transcardially 14-days post-implantation with 0.9% saline followed by 4% paraformaldehyde (in 0.2M PBS) to fix brain tissue before its removal from the skull. Brains were post-fixed with 4% paraformaldehyde for 24 h, then cryopreserved in 30% sucrose with sodium azide (Sigma-Aldrich, St. Louis, MO) at 4 °C. Brains were cut into 50 μm thick histological sections and placed directly onto microscopic slides to preserve tissue morphology.

#### 4.5.2. Immunohistochemistry

Brain tissue sections were washed 3x for 5 min with 0.01M PBS at room temperature (21 °C). Primary antibodies (Table 2) were diluted in PBS + 0.1% Triton X-100 (Sigma-Aldrich, St. Louis, MO) and applied to brain sections for overnight incubation without shaking (21 °C). Primary antibodies were washed off (3 × 5 min PBS), and secondary AlexaFluor 488, 555, 660 antibodies (1:1000 dilution) were applied for 1 h (21 °C). After removing the secondary antibodies, Hoechst (1 μg/mL, Sigma-Aldrich, St. Louis, MO) was applied for 1 min followed by 3 × 5 min washes with PBS. Finally, sections were coverslipped with fluorescent mounting medium (Dako, Carpinteria, CA) and stored at 4 °C prior to imaging. Visualization of antibodies was performed on a fluorescence microscope (Axioimager M2, Zeiss. Peabody, MA) interfaced with a monochrome camera driven by Stereo Investigator (MBF Bioscience, Williston, VT) image capture software [30].

#### 4.5.3. ECM Hydrogel Volume 

Whole hemisphere images were acquired using the virtual tissue module (MBF Bioscience, Williston, VT), which tiled individual 20× magnification images to create a composite image. Anterior-posterior images covered all slices containing the bioscaffold (500 μm apart). To calculate the total volume occupied by ECM hydrogel, the area occupied by the bioscaffold (i.e., collagen I staining) was multiplied by the distance between images. The degree of biodegradation was determined by dividing the ECM injection volume (i.e., MRI-based lesion volume) by the histology-based bioscaffold volume [19]. To establish the rate of biodegradation, the histology-based ECM volume was subtracted from the hydrogel injection volume before division by the number of days between injection and perfusion (i.e., 14 days). The degradation rate was defined as μL/day [19]. 

#### 4.5.4. Glial Scarring

Whole hemisphere images at 10× magnification were acquired stained for glial fibrillary acid protein (GFAP). Images were acquired with consistent 100 ms exposure time for all animals to measure signal intensity. Images were processed in Fiji software version 2.3.051, with measurement lines drawn from the cavity/lesion boundary to the subventricular zone. A plot of intensity versus distance determined the thickness of the glial scar. Starting from the lesion boundary, the average intensity of the GFAP signal was binned and averaged for every 200 μm interval. 

#### 4.5.5. Cell Invasion

The boundary between the extracellular matrix and the host tissue was delineated by the presence of collagen I [30]. The number and phenotype of cells within this area were subsequently described as cells that invaded the ECM [19,23]. Whole graft images were taken on slides with collagen I (to define the host-ECM boundary) and Hoechst (to identify cell nuclei) stains at 20× magnification. Images were processed through Fiji to quantify the number of invading cells. 

#### 4.5.6. Cell Phenotypes

Images were taken at 20× magnification within the ECM material, next to the area of damage, and on the contralateral hemisphere to determine the total number of cells (i.e., Hoechst+) and their phenotypes [19,23]. In all anterior-posterior slices containing ECM hydrogel, five to eight images were taken and counted at uniform distribution throughout the region-of-interest (ROI) within the bioscaffold, peri-infarct, and contralateral regions. Phenotypic markers for neural progenitor cells (doublecortin, DCX), neurons (NeuN), astrocytes (glial fibrillary acid protein, GFAP), oligodendrocytes (2′,3′-cyclic-nucleotide 3′-phosphodiesterase, CNPase), endothelial cells (rat endothelial cell antigen 1, RECA-1), macrophages (ionized calcium-binding adapter molecule-1, Iba-1), and their polarization (CD206 for M2 phenotype, CD86 for M1 phenotype) were used for analysis (Table 2). Phenotypes were expressed as cell density (i.e., cells/mm^3^).

### 4.6. Statistical Analyses 

Statistical analyses and graphing of data were performed in Prism 9 (Graph Pad, San Diego, CA). Two-way analyses of variance (ANOVA) were performed with significance set at *p* < 0.05. Each data point represents the mean value for each subject, whereas the group’s arithmetic mean is indicated by a horizontal line, and error bars reflect the standard deviation of data distribution. 

## 5. Conclusions

An acute stroke creates an evolving pathological microenvironment driven by the host inflammatory response [18]. Implantation of ECM hydrogel to induce brain tissue regeneration needs to be considered in the context of the local microenvironment encountered upon implantation. We here demonstrated that the formation of a tissue cavity is essential for the implantation of a bioscaffold aimed at replacing lost tissue. Optimal implantation timing was revealed to be around 28-days post-stroke, with a significantly reduced regeneration potential occurring by 90-days post-stroke. Nevertheless, studies with additional time points are desirable to more fully determine how particular microenvironmental factors (e.g., glial scar) affect the tissue regeneration process. The evolving host microenvironment determined the degree of the inflammatory response to the bioscaffold implantation, but the polarization of microglia/macrophages was defined by the ECM hydrogel. An insufficient neural progenitor invasion was noted, with neuron density being low. However, it remains unclear if a continued infiltration of neural progenitors will occur and how many neurons would be required in a de novo tissue to support behavioral recovery. The development of ECM hydrogel as a therapeutic strategy for stroke will need to address these issues to ensure that an appropriate tissue restoration can be guaranteed in each individual patient. 

## Figures and Tables

**Figure 1 ijms-22-11372-f001:**
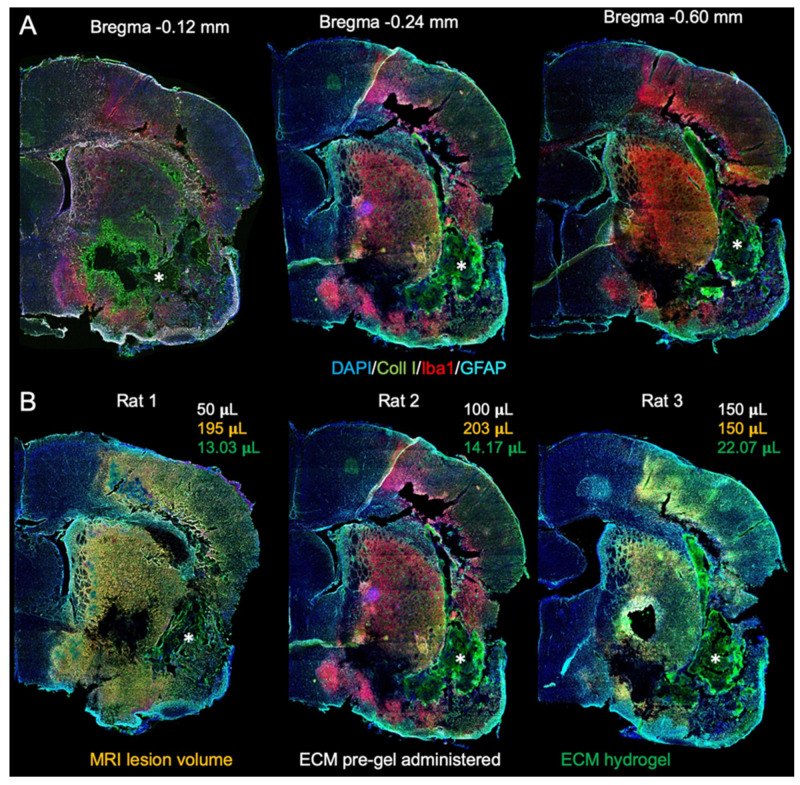
7-days post-stroke implantation of ECM hydrogel. (**A**) Macroscopic whole brain slice images were acquired to examine the distribution of ECM hydrogel injected 7-days post-stroke. Collagen I staining visualized the ECM hydrogel (*) based on an intensity difference of collagen I content in the bioscaffold compared to the veterate brain. (**B**) There was a poor distribution of ECM hydrogel throughout the stroke-damage region, irrespective of the administered pre-gel volume. Only a small volume (13–22 μL) of the administered ECM pre-gel was retained within these brains, akin to what is expected after intra-parenchymal implantations.

**Figure 2 ijms-22-11372-f002:**
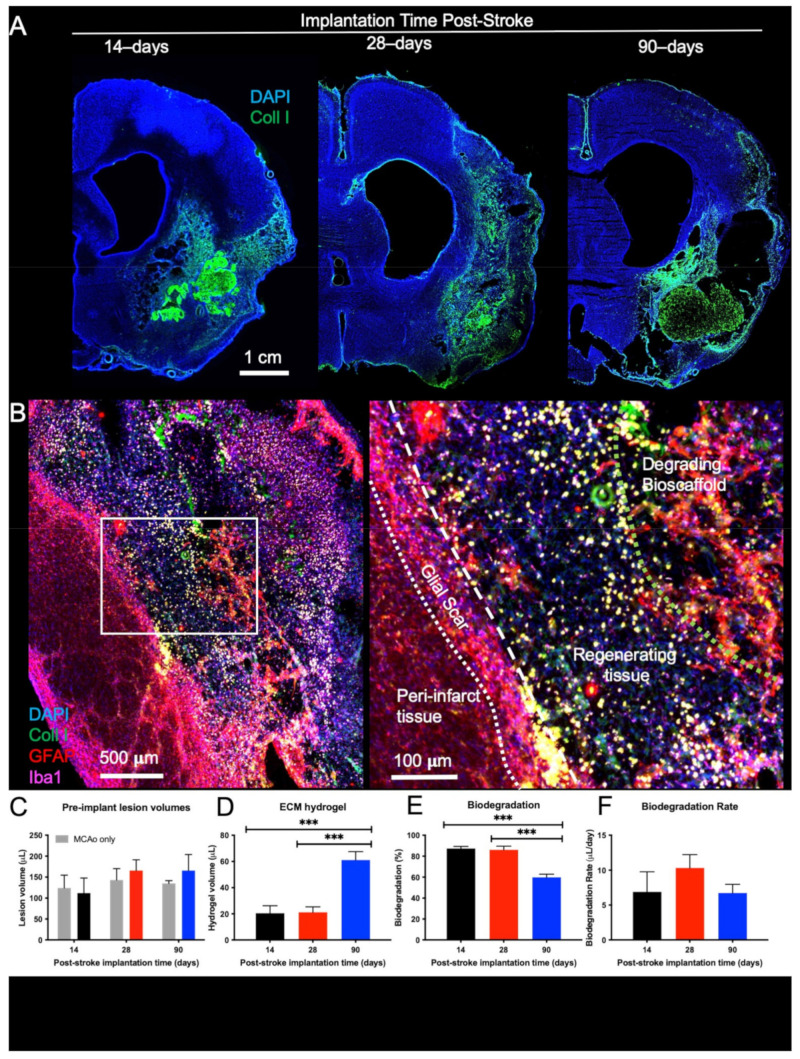
Macroscopic distribution and biodegradation of ECM hydrogel. (**A**) Macroscopic images were acquired to examine the distribution of the remaining ECM hydrogel 14-days after implantation. The 90-days post-stroke implantation condition shows limited biodegradation, while the 14- and 28-days injections revealed efficient biodegradation with only small volumes of hydrogel remaining. (**B**) A higher magnification image of the 28-days implantation illustrates the process of biodegradation. The boundary between host and lesion cavity is defined by the glial scar (GFAP); therefore, the cells (DAPI) present within this lesion are considered regenerating tissue. The remaining bioscaffold (Collagen I) can be seen surrounded by astrocytes (GFAP) and macrophages (Iba-1). (**C**) Lesion volumes were calculated from T_2_-weighted MR images acquired 10-days post-stroke. Random group assignment of rats resulted in equivalent lesion volumes in all conditions. (**D**) The amount of remaining ECM hydrogel was quantified to determine the biodegradation for the 14-, 28-, and 90-days post-stroke implantations. (**E**) The 14- and 28-days post-stroke implants had no significant difference in the percentage of biodegradation, while the 90-days post-stroke implantation condition demonstrated significantly less biodegradation (*** *p* < 0.001). (**F**) However, the degradation rate (μL/day) was not significantly different between time points.

**Figure 3 ijms-22-11372-f003:**
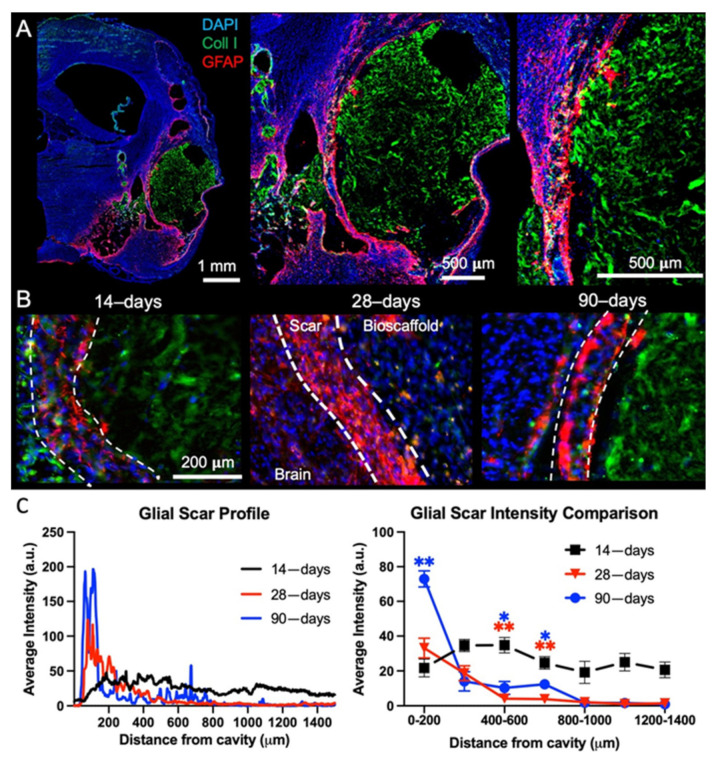
Glial scarring and tissue astrocytosis. (**A**) To assess the impact of ECM hydrogel implantation timing on glial scarring, whole hemisphere images encompassing the lesion cavity were acquired to measure the level of astrocytic (GFAP) reactivity at each time point. A 90-days post-stroke implantation brain is shown here. (**B**) It is important to note the morphological difference in astrocytic activity at different time points, with the 14-days post-stroke implantation exhibiting more diffuse scarring, whereas 90-days post-stroke, the glial scar has become more compact. In the case of the 28-days post-stroke implantation, the ECM hydrogel adjacent to the glial scar was degraded and replaced with de novo tissue, as evidenced here by the almost complete absence of collagen I staining on the lesion cavity/bioscaffold side. (**C**) A quantitative analysis of the scar profile confirms an intensity increase in GFAP from the 14- to 90-days post-stroke at the edge of the lesion. In contrast, a decrease of astrocytosis away from the cavity border was evident at 90-days compared to 14-days post-stroke. (* *p* < 0.01, ** *p* < 0.001)

**Figure 4 ijms-22-11372-f004:**
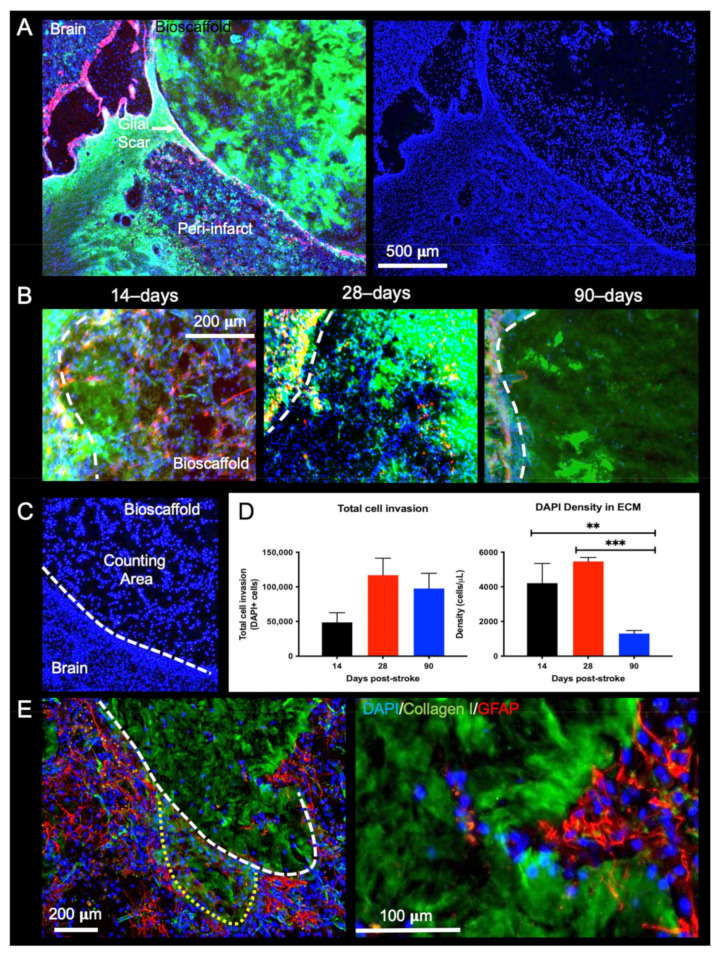
Presence of host cells in ECM hydrogel. (**A**) Using collagen I staining, a region of interest (ROI) was defined around the edge of the biomaterial, which coincides with the presence of a glial scar (GFAP). The 14-days post-stroke implantation condition is shown here. DAPI staining was overlaid to visualize the presence of cells within the hydrogel. (**B**) The 14- and 28-days post-stroke implants resulted in a large number of DAPI cells within the ROI, whereas a low density of cells was evident in the 90-days post-stroke condition. (**C**) In order to quantify the number of cells within the lesion, DAPI+ cells were counted within the ROI. (**D**) No significant difference was found between time points for total cell invasion. However, cell density quantification indicated a significantly lower cell density in bioscaffolds implanted 90-days post-stroke (** *p* < 0.01, *** *p* < 0.001). Cell density in bioscaffolds implanted 14- and 28-days post-stroke were more than twice as those in the 90-days condition. (**E**) In the 14-days post-stroke condition, the peri-infarct tissue consisted of a more diffuse GFAP+ cell distribution facilitating the infiltration of host cells forming a macro track into the bioscaffold.

**Figure 5 ijms-22-11372-f005:**
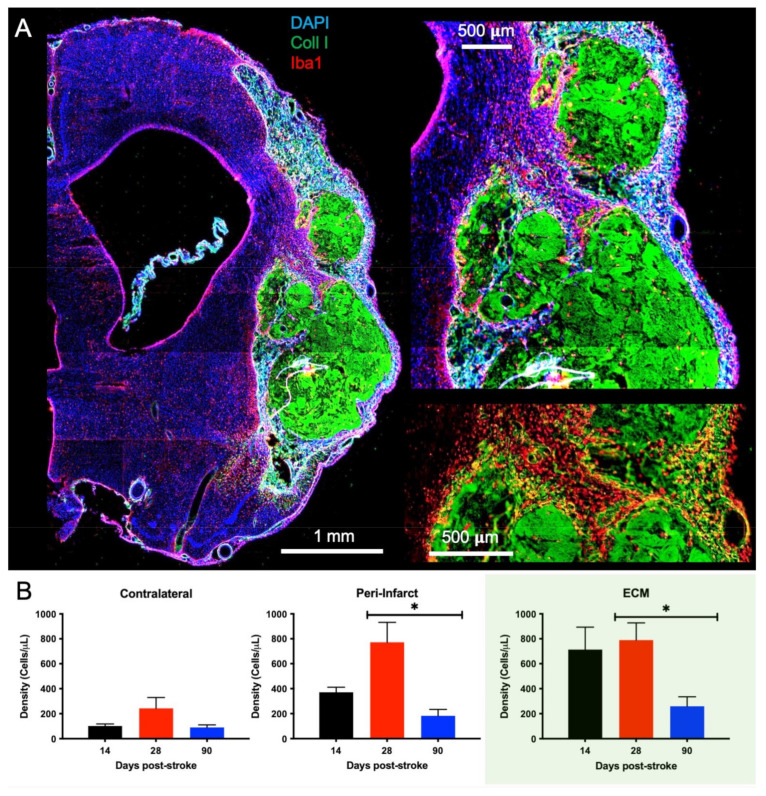
Macrophage Invasion. (**A**) A whole hemisphere image reveals the accumulation of Iba1+ cells around the implanted ECM hydrogel (collagen I+). A magnified view of the area between the cortex and striatum further highlights the invasion of Iba-1+ macrophage is evident at the tissue/hydrogel interface (90-days post-stroke implant). Visualization of only the Iba-1+ cells and the ECM hydrogel (bottom right image) accentuates the dense accumulation of macrophages in tissue before infiltrating the bioscaffold. (**B**) Quantification of the Iba-1+ cells revealed a much larger density of macrophages within and surrounding the ECM compared to the contralateral hemisphere. The 90-days post-stroke implantation also showed the least macrophage infiltration compared to the 14- and 28-days time points (* *p* < 0.05), with <500 cells/μL within the material.

**Figure 6 ijms-22-11372-f006:**
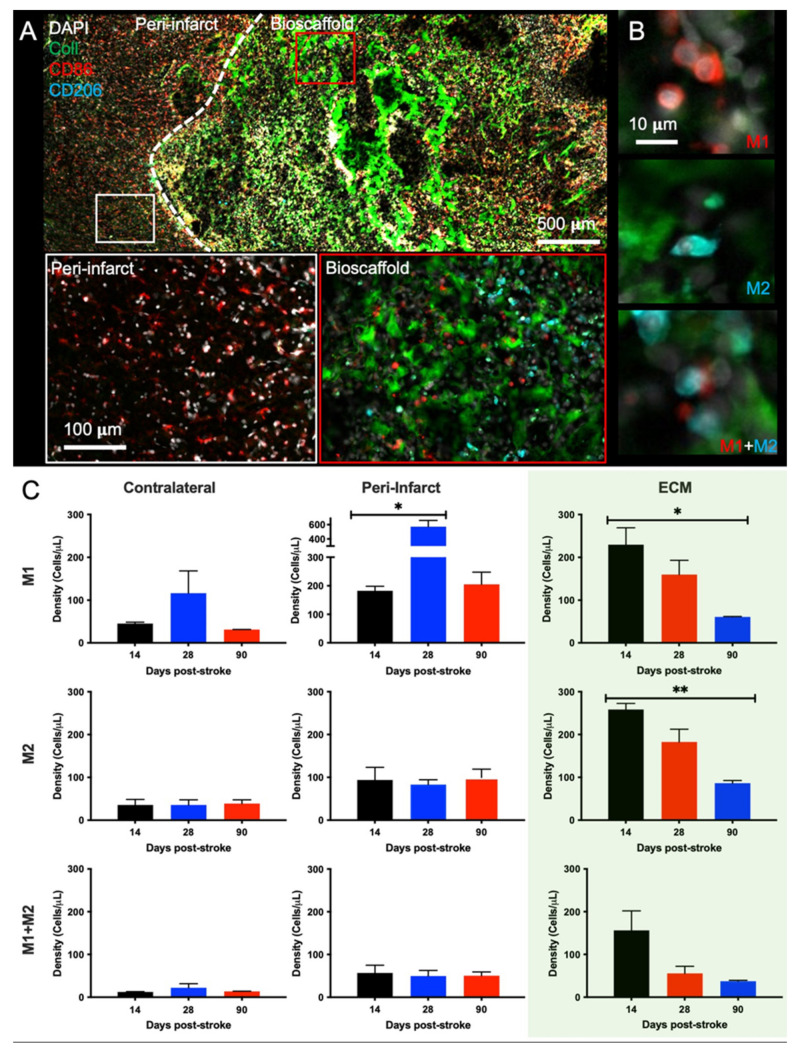
Phenotypic characterization of invading immune cells in ECM hydrogel. (**A**) An invasion of M1- (CD86+) and M2-like (CD206+) macrophages is evident both within the ECM hydrogel and the surrounding peri-infarct region (28-days post-stroke condition). In the peri-infarct region, there is a much larger proportion of M1- to M2-like macrophages compared to within the bioscaffold. (**B**) Within the bioscaffold, M1-like and M2-like polarization of macrophages was evident, with some cells expressing both markers. (**C**) Quantification of the M1/M2 polarization revealed an almost 1:1 ratio of M1- to M2-like macrophages across all time points within the bioscaffold. In the peri-infarct regions, there was a high concentration of M1-like macrophages, especially the 28-days post-stroke, while M2 and M1+M2 macrophage levels remained low. In the contralateral hemisphere, very few macrophages were detected. (* *p* < 0.05; ** *p* < 0.01).

**Figure 7 ijms-22-11372-f007:**
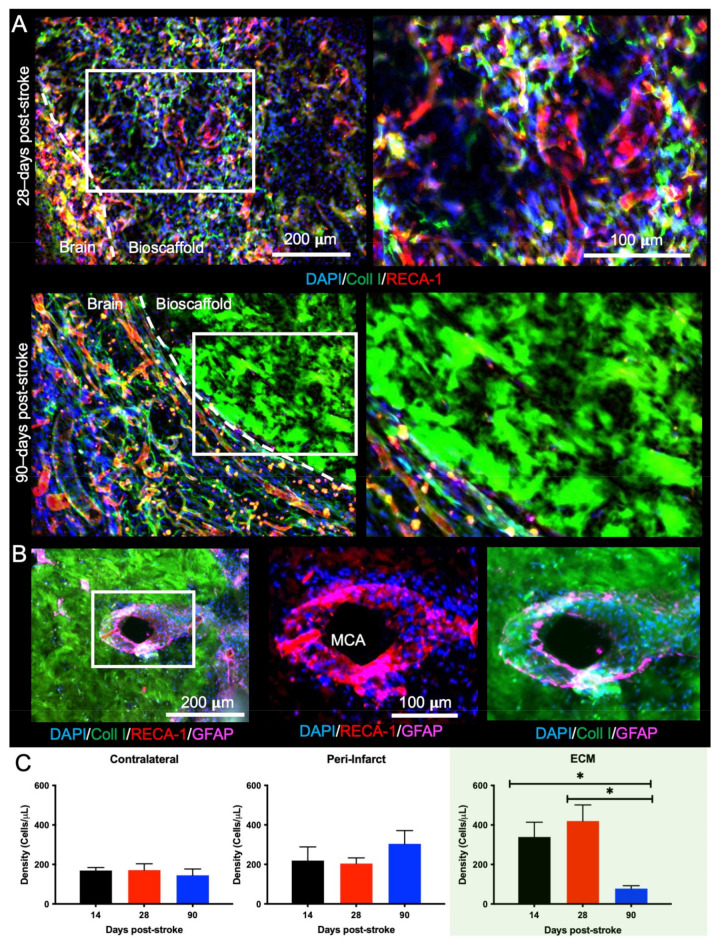
Neovascularization of the ECM hydrogel. (**A**) Neovascularization inside the hydrogel was evident if the ECM hydrogel underwent sufficient biodegradation, as illustrated here when the material was implanted 28-days post-stroke. However, there was a lack of vascular formation when the bioscaffold was implanted 90-days post-stroke, despite the number of blood vessels observed in the peri-infarct region. (**B**) The remnant of the middle cerebral artery (MCA) was surrounded by cells invading into the bioscaffold implanted 90-days post-stroke, illustrating one method of cell infiltration is for host cells to migrate along blood vessels. (**C**) A quantification of RECA-1+ cells indicated that the 90-days post-stroke condition had the lowest infiltration of endothelial cells (ECs) compared to the 14- and 28-days’ time points. The peri-infarct region presented with a slightly higher density of endothelial cells than the contralateral hemisphere across all time points (* *p* < 0.05).

**Figure 8 ijms-22-11372-f008:**
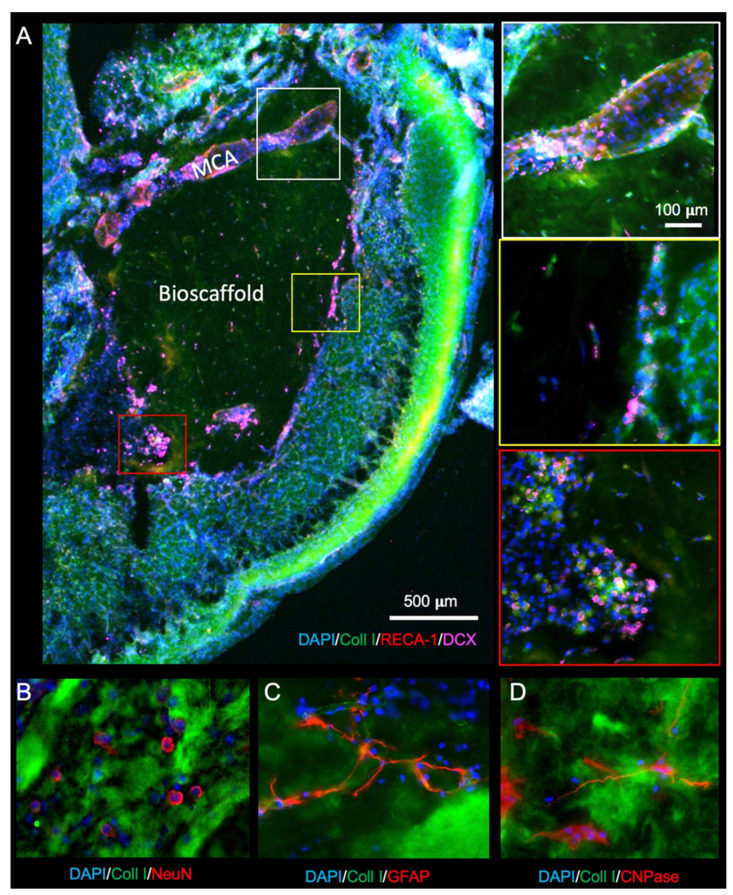
Neuronal and glial cell invasion into the ECM hydrogel. (**A**) Neural progenitor cells (doublecortin, DCX) are observed infiltrating the biomaterial (28-days post-stroke) along the middle cerebral artery (MCA), as well as in groups at the edge of the material. (**B**) To determine whether mature neurons were differentiating in de novo tissue, NeuN staining was performed to target post-mitotic neurons. (**C**) Astrocytes (GFAP+ cells), which form the glial scar surrounding the damaged tissue, presented with long processes extending through the ECM hydrogel. (**D**) Oligodendrocytes (CNPase) also infiltrated the bioscaffolds in the 14- and 28-day post-stroke conditions but less so 90-days post-stroke.

**Figure 9 ijms-22-11372-f009:**
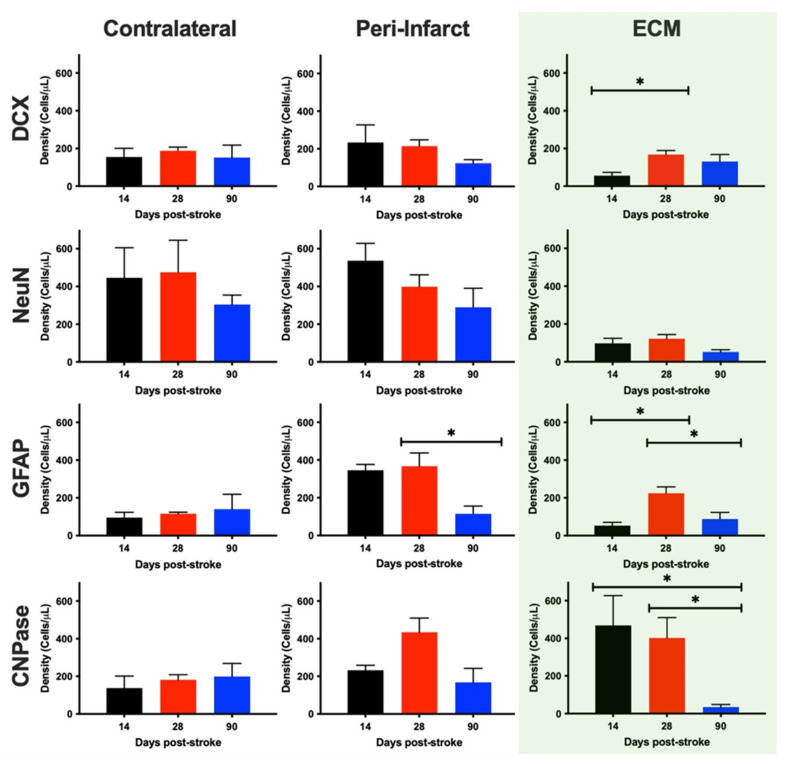
Phenotypic characterization and quantification of invading neural cells. To provide an anatomical context of cells within the bioscaffold, the same phenotypes were quantified in the contralateral and peri-infarct regions. To afford a comparison across different ECM hydrogel volumes, cell densities (cells/μL) were calculated. Within the bioscaffold, neural progenitors (DCX+ cells) infiltrated across all time points with the 28-days post-stroke time point yielding the highest density of progenitors, comparable to the levels found in the contralateral and peri-infarct regions. In contrast, mature neurons (NeuN+) were dramatically reduced in the bioscaffold, but there was no difference in content based on when the ECM hydrogel was implanted. Astrocytes (GFAP+) exhibited the greatest density in peri-infarct regions. Within the ECM hydrogel, a peak astrocyte content was evident in the 28-days post-stroke condition. Oligodendrocyte (CNPase+) density was increased in the peri-infarct region and was highest within the bioscaffold if these were implanted between 14- and 28-days post-stroke. An almost complete lack of oligodendrocytes with the ECM hydrogel occurred in the 90-days post-stroke condition. (* *p* < 0.05).

**Table 1 ijms-22-11372-t001:** Number of animals assigned to experimental condition.

Conditions	Day 14	Day 28	Day 90
MCAo only	5	5	5
MCAo + ECM	5	5	4

**Table 2 ijms-22-11372-t002:** List of antibodies used for immunohistochemistry.

Antibody	Concentration	Company	Cat. Ref.	Clone
Collagen-I	1:250	Abcam	Ab34710	Collagen I aa 1-1464
Iba-1	1:300	Abcam	Ab5076	Iba1 aa 135-147
GFAP	1:3000	Sigma	G3893	G-A-5
DCX	1:150	Abcam	Ab153668	CAA0661′7.1 and AAT58219.1
CNPase	1:200	Abcam	Ab6319	11-5B
Fox3	1:500	Abcam	Ab104224	1B7
RECA-1	1:100	Abcam	Ab9774	RECA-1
CD86	1:200	Abcam	Ab53004	EP1159Y
CD206	1:200	Santa Cruz	sc-34577	C-20

## Data Availability

Images and data are available from the corresponding author upon reasonable request.

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
