# Peer review of "Post-Stroke Timing of ECM Hydrogel Implantation Affects Biodegradation and Tissue Restoration"

_ijms, 2021, doi:10.3390/ijms222111372_

Round 1

Reviewer 1 Report

It is a well-structured article, the purpose of the study is well exemplified and the methods used are described in detail and explicitly. The results are valuable and represent a continuation of studies started a few years ago by members of the same team of authors. The discussions implement these results in the current context of studies on post-stroke tissue restoration methods based on the use of ECM hydrogel. However, I made some minor observations on certain figures, which can be seen in the Adobe pdf format of the article (Open with Adobe). 

Author Response

  1. Page 2. It is necessary to specify the animal model that was worked on.

We have now added “rats” to the sentence highlighted by the reviewer.

  1. Page 10. Figure 2B needs to be better explained, especially the 28-day image which is very different from the 14- and 90-day images. No bioscaffold collagen is observed. If the bioscaffold underwent a degradation, this must be mentioned.

We have amended the description of Figure 2B to indicate that the image for the 28 day time points showing that “ECM hydrogel adjacent to the glial scar was degraded and replaced with de novo tissue, as evidenced here by the almost complete absence of collagen I staining on the lesion cavity/bioscaffold side.”

  1. Page 13. Figure 4A. An explanation is required for the image at the bottom right A (without DAPI). Although it is understood what the image represents, the difference from the image above must be specified.

We have now revised the description of this part of the Figure to more clearly describe its different constituent elements, as well as specifically mentioning the bottom right image.

  1. Page 16: Figure 5B shows nothing but a few blurry colors. Two red spots and a blue spot prove nothing. The enlarged and clear images of figure 5A would be much more explicit, highlighting the enlarged area.

We beg to respectfully disagree with the reviewer. Images in 5A show population dynamics of polarized macrophages which is require to highlight here what has not been previously documented that all macrophages in peri-infarct tissue are of a M1 phenotype, whereas within the ECM hydrogel we see a shift in polarization with M1 and M2 phenotypes present. Higher magnification images of this can be provided but these images are essential to demonstrate the differential tissue distribution of these phenotypes. We very much disagree that Figure 5B only shows blurred spots proving nothing! These images clearly show the differential polarization of macrophages within the ECM hydrogel, as well as the shift of these from a M1 to a M2 phenotype (M1+M2 image), which is key to their quantification reported in 5C.

  1. Page 17: Figure 6B. The explanations show "middle cerebral artery" (MCA), which does not appear in the image. Each image should explain the colors, so the collagen, green, is visible only in the first image, not in the second (the third image is the magnification of the second).

We have now replaced “blood vessel” with MCA on the image to be consistent. We have added a box to the first image to show the zoomed in area of the MCA. We replaced the middle image with the zoom overlay of RECA1 and DAPI to emphasizes the infiltration of host cells through the vessel wall into the ECM hydrogel. We have now positioned the labels for this combination under the image to avoid any confusion between images. A 3rd image was added of the zoomed in area to show the distribution of DAPI cells against the ECM hydrogel. We generated the separate image combination to allow the reader to more clearly see how these different components interact, which is not as easily visible when all channels are combined.

Reviewer 2 Report

  1. Introduction and abstract section is well written with appropriate references and background information on need of ECM hydrogels. However, can authors detail the reasons why certain timepoints were chosen and if there is a room for more timepoint study.
  2. Methods section is detailed and contains all the information needed. Authors have paid a great attention to detail in providing the details if experiments need to be duplicated. 
  3. In methods section, can authors specify type/model of Fiji?
  4. In section 3.1, can authors detail on the significance of choosing 50/100 and 150um of ECM hydrogel?
  5. Can authors explain what sort of glial scarring and tissue regeneration will be expected between 28 and 90 days time points? Since we see stark difference between 28 and 90 days? 
  6. Results/discussion and conclusion is well explained for readers. 

Author Response

  1. Introduction and abstract section is well written with appropriate references and background information on need of ECM hydrogels. However, can authors detail the reasons why certain timepoints were chosen and if there is a room for more timepoint study.

We have amended method section 2.4 to indicate the rationales for using the given time points based on the stroke lesion evolution. We further amended the conclusion to indicate that additional time points are desirable to more fully determine how particular microenvironmental factors affect the tissue regeneration process.

  1. Methods section is detailed and contains all the information needed. Authors have paid a great attention to detail in providing the details if experiments need to be duplicated.

Thank you!

  1. In methods section, can authors specify type/model of FIJI?

We have now updated this information to indicate version 2.3.051 was used.

  1. In section 3.1, can authors detail on the significance of choosing 50/100 and 150 um of ECM hydrogel?

We have now corrected to indicate that these represent escalating volumes to determine if this allows the delivery of a greater volume to this region. We have further amended this section to indicate that the presence of tissue remnants prevented the delivery of bioscaffold volumes required to cover the lesion area.

  1. Can authors explain what sort of glial scarring and tissue regeneration will be expected between 28 and 90 das time points? Since we see stark difference between 28 and 90 days?

We agree with the reviewer, this is a key question that emerges from this work. We have amended section 4.1 to indicate that it remains unclear what matural changes in glial scarring occur when between these two time points that would shift the endogenous tissue response from a permissible to a prohibitive milieu. This is certainly a major question that requires a detailed experiment that would characterize different aspects of scarring across multiple time points between these two extreme responses and ideally try to manipulate the scarring process to conclusively demonstrate a mechanistic switch.

  1. Results/Discussion and Conclusion is well explained for readers.

Thank you!